

# The Non-Recurrent Laryngeal Nerve: a meta-analysis and clinical considerations

Brandon Michael Henry[1,2], Silvia Sanna[3], Matthew J. Graves[1,2], Jens Vikse[2,4], Beatrice Sanna[5], Iwona M. Tomaszewska[6], R. Shane Tubbs[7], Jerzy A. Walocha[1,2] and Krzysztof A. Tomaszewski[1,2]

[1] Department of Anatomy, Jagiellonian University Medical College, Kraków, Poland
[2] International Evidence-Based Anatomy Working Group, Kraków, Poland
[3] Department of Surgical Sciences, University of Cagliari, Monserrato, Sardinia, Italy
[4] Division of Medicine, Stavanger University Hospital, Stavanger, Norway
[5] Faculty of Medicine and Surgery, University of Cagliari, Monserrato, Sardinia, Italy
[6] Department of Medical Education, Jagiellonian University Medical College, Kraków, Poland
[7] Seattle Science Foundation, Seattle, WA, United States

Corresponding author
Brandon Michael Henry,
bmhenry55@gmail.com

## ABSTRACT

**Background**. The Non-Recurrent Laryngeal Nerve (NRLN) is a rare embryologically-derived variant of the Recurrent Laryngeal Nerve (RLN). The presence of an NRLN significantly increases the risk of iatrogenic injury and operative complications. Our aim was to provide a comprehensive meta-analysis of the overall prevalence of the NRLN, its origin, and its association with an aberrant subclavian artery.

**Methods**. Through March 2016, a database search was performed of PubMed, CNKI, ScienceDirect, EMBASE, BIOSIS, SciELO, and Web of Science. The references in the included articles were also extensively searched. At least two reviewers judged eligibility and assessed and extracted articles. MetaXL was used for analysis, with all pooled prevalence rates calculated using a random effects model. Heterogeneity among the included studies was assessed using the Chi$^2$ test and the I$^2$ statistic.

**Results**. Fifty-three studies (33,571 right RLNs) reported data on the prevalence of a right NRLN. The pooled prevalence estimate was 0.7% (95% CI [0.6–0.9]). The NRLN was found to originate from the vagus nerve at or above the laryngotracheal junction in 58.3% and below it in 41.7%. A right NRLN was associated with an aberrant subclavian artery in 86.7% of cases.

**Conclusion**. The NRLN is a rare yet very clinically relevant structure for surgeons and is associated with increased risk of iatrogenic injury, most often leading to temporary or permanent vocal cord paralysis. A thorough understanding of the prevalence, origin, and associated pathologies is vital for preventing injuries and complications.

## INTRODUCTION

The Non-Recurrent Laryngeal Nerve (NRLN) is a rare variant of the Recurrent Laryngeal Nerve (RLN) that takes an aberrant course, not descending into the thorax as is usual (Fig. 1). It was first reported by *Stedman (1823)*. It arises almost exclusively on the right
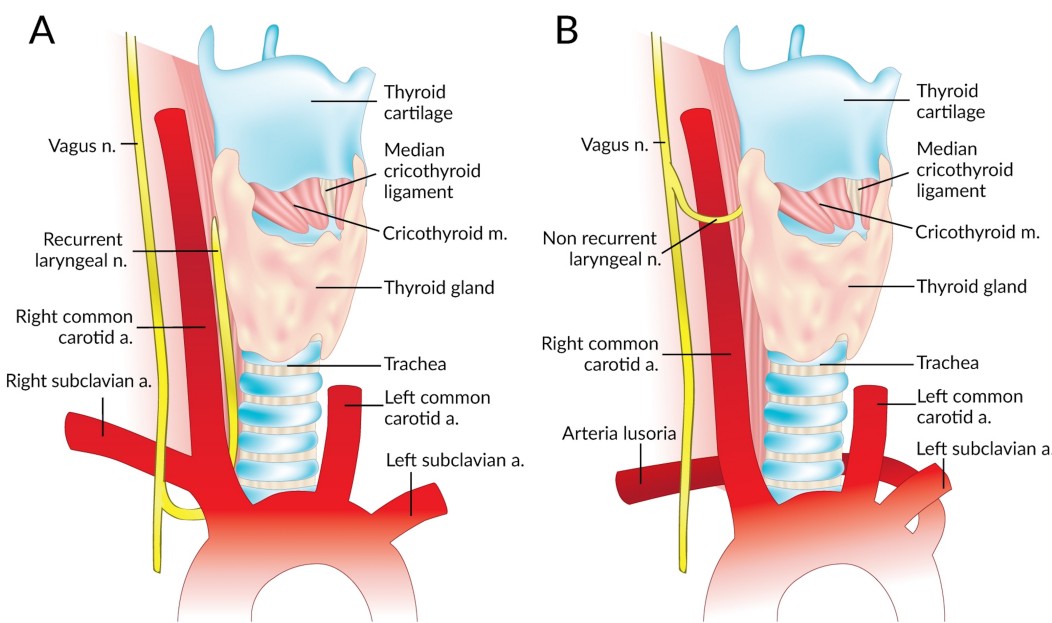

**Figure 1 Normal right recurrent laryngeal nerve (A) and right non-recurrent laryngeal nerve in the presence of an aberrant subclavian artery (B).**

side and is closely associated with vascular anomalies of the aortic arch (*Wang et al., 2011*). On the right side, the NRLN usually results from partial regression of the fourth pharyngeal arch, resulting in an aberrant subclavian artery (arteria lusoria) that runs behind the esophagus (Fig. 1B) (*Wang et al., 2011*). This atypical vascular pattern permits the nerve to migrate freely into the neck as the fetus grows longitudinally (*Wang et al., 2011*). The existence of an NRLN without the associated vascular anomalies has no clear embryological explanation (*Wang et al., 2011*). NRLNs on the left side have only been reported a few times, all of them accompanied by other significant pathologies such as situs inversus (*Henry et al., 1988*; *Toniato et al., 2004*; *Hong, Park & Yang, 2014*).

The NRLN variant of the RLN is a major risk factor for iatrogenic injury and can lead to detrimental postoperative complications if its existence is not observed in a timely fashion. As noted by *Toniato et al. (2004)*, patients experienced a nearly six-fold increase in intraoperative nerve injuries if they had an undetected NRLN. A thorough dissection in all approaches to thyroidectomy, parathyroidectomy, and endarterectomy is essential for identifying the neurovascular structures and preventing intra- and post-operative nerve complications, the most common of which is vocal cord paralysis (*De Luca et al., 2000*; *Hong, Park & Yang, 2014*). It was noted in *Iacobone et al. (2015)* that preoperative ultrasonography (USG) to assess patients for an NRLN was extremely successful, with an accuracy of more than 98%. It is therefore strongly suggested that measures such as preoperative USG are taken to identify these variant structures because they clearly help to prevent injury. In Iacobone's study, nerve palsy did not arise in the ultrasound group

yet arose 3 times in the control group, a true testament to the importance of preoperative identification (*Iacobone et al., 2015*).

The prevalence of the NRLN has been reported numerous times with rates ranging from 0% to 4.76% (*Menck, Grüber & Lierse, 1990*; *Freschi et al., 1994*; *Moreau et al., 1998*; *Sasou, Nakamura & Kurihara, 1998*; *Sturniolo et al., 1999*; *Monfared, Gorti & Kim, 2002*; *Page, Foulon & Strunski, 2003*; *Makay et al., 2008*; *Lee et al., 2009*; *Kandil et al., 2011*; *Benouaich et al., 2012*; *Ngo Nyeki et al., 2015*). It is essential to obtain accurate anatomical data on the NRLN if patients with this anomaly are to be assessed properly for surgical candidacy and operative planning. The aim of our analysis was to provide a comprehensive and evidence-based assessment of the prevalence of the NRLN. We also aimed to investigate the course-related consequences of the different types of NRLN, and the association of this variant nerve with the incidence of an aberrant subclavian artery. Since the RLN, and in particular the NRLN, are particularly susceptible to surgical injury, a complete understanding and assessment of their variant anatomy is essential for preventing injuries and ensuring complication-free procedures.

## METHODS

### Search strategy

Through March 2016, a database search was performed of PubMed, CNKI, ScienceDirect, EMBASE, BIOSIS, SciELO, and Web of Science in order to identify eligible articles for the meta-analysis. The exhaustive search strategy employed for PubMed is presented in Table S1. No date limits or language restrictions were applied. The references in the included articles were also extensively searched. The Preferred Reporting Items for Systematic Reviews and Meta-Analyses (PRISMA) guidelines were strictly followed throughout this meta-analysis (Table S2) (*Moher et al., 2009*). We prospectively registered the meta-analysis in PROSPERO (CRD42015026096).

### Criteria for study selection

Studies were considered eligible for inclusion in the meta-analysis if they: (1) reported clear, extractable prevalence data on the non-recurrent laryngeal nerve with respect to side and (2) were cadaveric, intraoperative, or imaging studies. The exclusion criteria included: (1) case studies, case reports, conference abstracts, and letters to the editor; (2) studies reporting incomplete data (i.e., not reporting rates with respect to side); and (3) studies on patients with trauma to the head and neck region. The decision to include only articles reporting rates with respect to side was based on the previously-established difference in prevalence rates of the NRLN between the right and the left sides (*Henry et al., 1988*).

All studies were independently assessed for eligibility by three reviewers (SS, JV and BS). Any disparities arising during the assessment were resolved by a consensus among all the reviewers, after consulting with the authors of the original study, if possible. All full-text articles published in languages not spoken fluently by the authors were translated for further eligibility assessment by medical professionals fluent in both English and the original language of the manuscript.

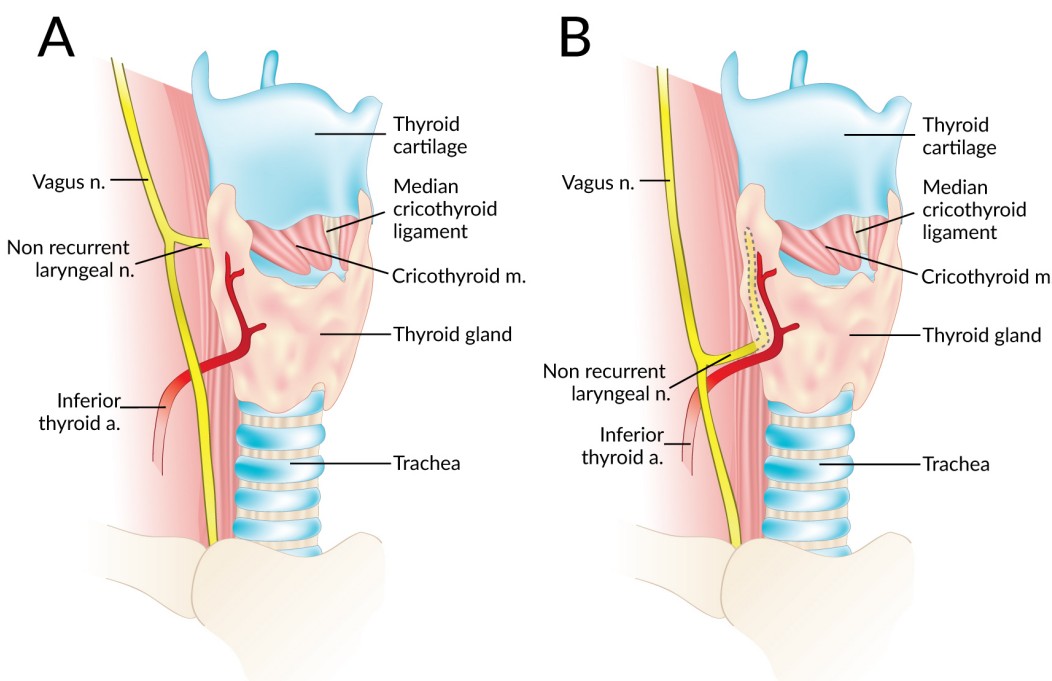

**Figure 2** A non-recurrent laryngeal nerve originating from the vagus nerve above (A) and below (B) the laryngotracheal junction.

## Data extraction

Data were independently extracted from the included articles by two independent reviewers (SS and JV). These data included demographic information such as year, country, type of study, study design, and number of nerves. The primary outcome, the prevalence of right and left NRLNs was isolated. Secondary outcomes such as the level of origin of the NRLN from the vagus nerve (at or above the level of the laryngotracheal junction or below that level) (Fig. 2), and the prevalence of an aberrant subclavian artery when an NRLN was present were also noted when available. In the event of data inconsistencies, the reviewers attempted to contact the authors of the original study by email for clarification.

## Statistical analysis

Statistical analysis was performed by BMH and MG using MetaXL version 2.0 by EpiGear International Pty Ltd (Wilston, Queensland, Australia). All pooled prevalence rates were calculated using a random effects model (*Henry, Tomaszewski & Walocha, 2016*). The Chi$^2$ test and the $I^2$ statistic were used to measure the level of heterogeneity among the included studies. For the Chi$^2$ test, a Cochran's Q *p*-value of <0.10 indicated significant heterogeneity (*Higgins & Green, 2011*). The values of the $I^2$ statistic were interpreted as follows: 0–40% might not be important; 30–60% could indicate moderate heterogeneity; 50–90% could indicate substantial heterogeneity; and 75–100% indicated considerable heterogeneity (*Higgins & Green, 2011*).

To probe the etiology of heterogeneity, subgroup analysis was performed on the basis of type of study (cadaveric vs. intraoperative), study design (prospective vs. retrospective),

and geographical origin of the articles. Significant differences between subgroups were judged from the confidence intervals of the rates, any overlap between groups indicating a lack of statistical significance (*Henry, Tomaszewski & Walocha, 2016*). Furthermore, a leave-one-out sensitivity analysis was performed to explore the source of heterogeneity.

## RESULTS

### Study identification

Figure 3 presents an overview of the flow of studies in the meta-analysis. Through database searching, 2,795 initial articles were identified. A further 84 articles were identified from reference searching. After removing duplicates and primary screening, 328 articles were assessed by full text for eligibility in the meta-analysis. Of these, 53 were deemed eligible and included, while 275 were excluded, 21 for not reporting extractable NRLN rates with respect to side.

### Characteristics of included studies

The characteristics of the studies included in the meta-analysis are summarized in Table 1, along with the reported prevalence of a right NRLN. A total of 53 studies ($n = 53,577$ total nerves; 33,571 Right RLNs and 20,006 Left RLNs) were included: 35 intraoperative, 17 cadaveric and 1 imaging (CT) (*Reed, 1943*; *Wade, 1955*; *Hunt, Poole & Reeve, 1968*; *Stewart, Mountain & Colcock, 1972*; *Skandalakis et al., 1976*; *Papadatos, 1978*; *Proye et al., 1982*; *Flament, Delattre & Palot, 1983*; *Henry et al., 1988*; *Menck, Grüber & Lierse, 1990*; *Lekacos et al., 1992*; *Freschi et al., 1994*; *Moreau et al., 1998*; *Sasou, Nakamura & Kurihara, 1998*; *Sturniolo et al., 1999*; *Campos & Henriques, 2000*; *Raffaelli, Iacobone & Henry, 2000*; *Watanabe et al., 2001*; *Watanabe et al., 2016*; *Monfared, Gorti & Kim, 2002*; *Page, Foulon & Strunski, 2003*; *Hermans et al., 2003*; *Ardito et al., 2004*; *Toniato et al., 2004*; *Spartà et al., 2004*; *Sciumè et al., 2005*; *Shindo, Wu & Park, 2005*; *Beneragama & Serpell, 2006*; *Maranillo et al., 2008*; *Makay et al., 2008*; *Serpell, Yeung & Grodski, 2009*; *Lee et al., 2009*; *Sunanda, Tilakeratne & De Silva, 2010*; *Shao et al., 2010*; *Wang et al., 2011*; *Kaisha, Wobenjo & Saidi, 2011*; *Kandil et al., 2011*; *Pradeep, Jayashree & Harshita, 2012*; *Chiang et al., 2012*; *Tang et al., 2012*; *Benouaich et al., 2012*; *Asgharpour et al., 2012*; *Silva, Siqueira & Arruda, 2013*; *Donatini, Carnaille & Dionigi, 2013*; *Satoh et al., 2013*; *Cai et al., 2013*; *Hong, Park & Yang, 2014*; *Yang et al., 2014*; *Han, Bai & Lu, 2015*; *Dolezel et al., 2015*; *Iacobone et al., 2015*; *Buła et al., 2015*; *Ngo Nyeki et al., 2015*; *Barczyński et al., 2015*). The dates of the included studies spanned the period from 1943 to 2016. Their geographical distribution was extremely wide, the most substantial contributions coming from Europe (27 studies) and Asia (15 studies).

### Prevalence of a right NRLN

A total of 53 studies (33,571 right RLNs) reported data on the prevalence of a right NRLN. The overall pooled prevalence estimate in the general population was 0.7% (95% CI [0.6–0.9]; $I^2 = 42.5$ (95% CI [20.2–58.6]); $p = 0.001$) (Fig. 4).

In subgroup analysis, the pooled prevalence of a right NRLN was significantly higher in cadaveric (1.4%, 95% CI [0.9–2.0]) than intraoperative (0.7%, 95% CI [0.5–0.8]) studies (Table 2). Subgroup analysis by geographical origin revealed no significant differences

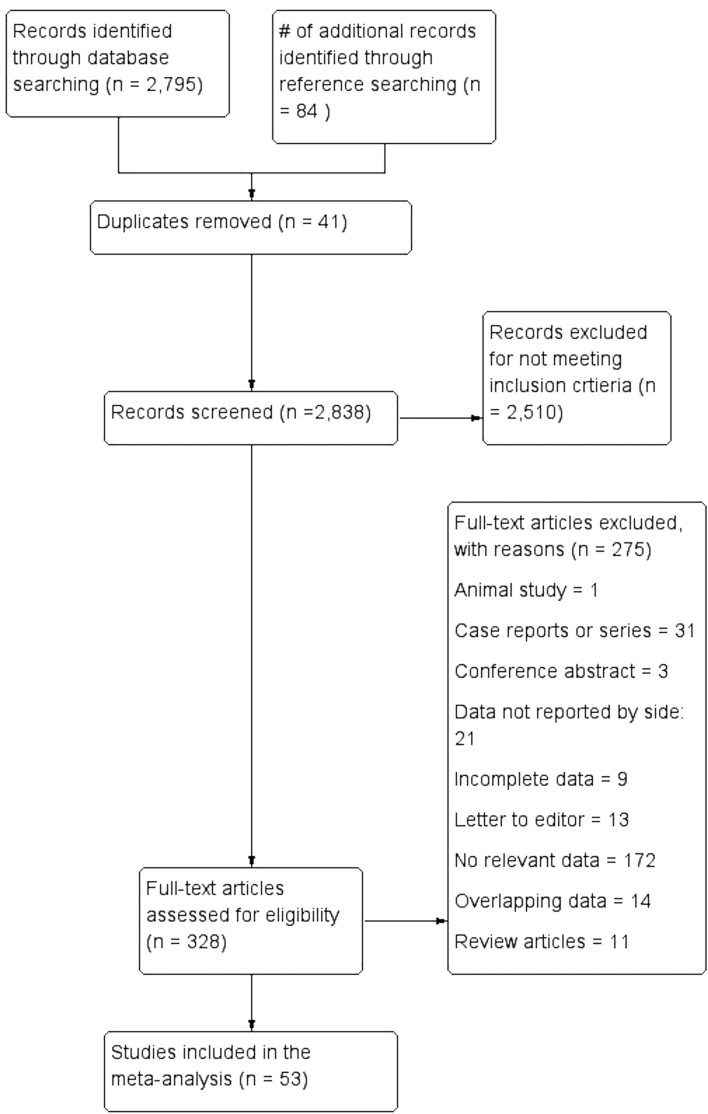

**Figure 3 Flowchart of study search, eligibility assessment, and inclusion.**

(Table 3), and no significant differences were detected in the leave-one-out sensitivity analysis.

## Types of right NRLN

A total of 14 studies ($n = 81$ right NRLNs) reported extractable data on the type of NRLN with respect to its level of origin from the vagus nerve. In 58.3% (95% CI [36.1–79.0]) of cases, the NRLN originated at or above the level of the laryngotracheal junction, while in 41.7% (95% CI [21.0–63.9]) it originated below that level ($I^2 = 67.6\%$, 95% CI [43.4–81.4]; $p < 0.001$).

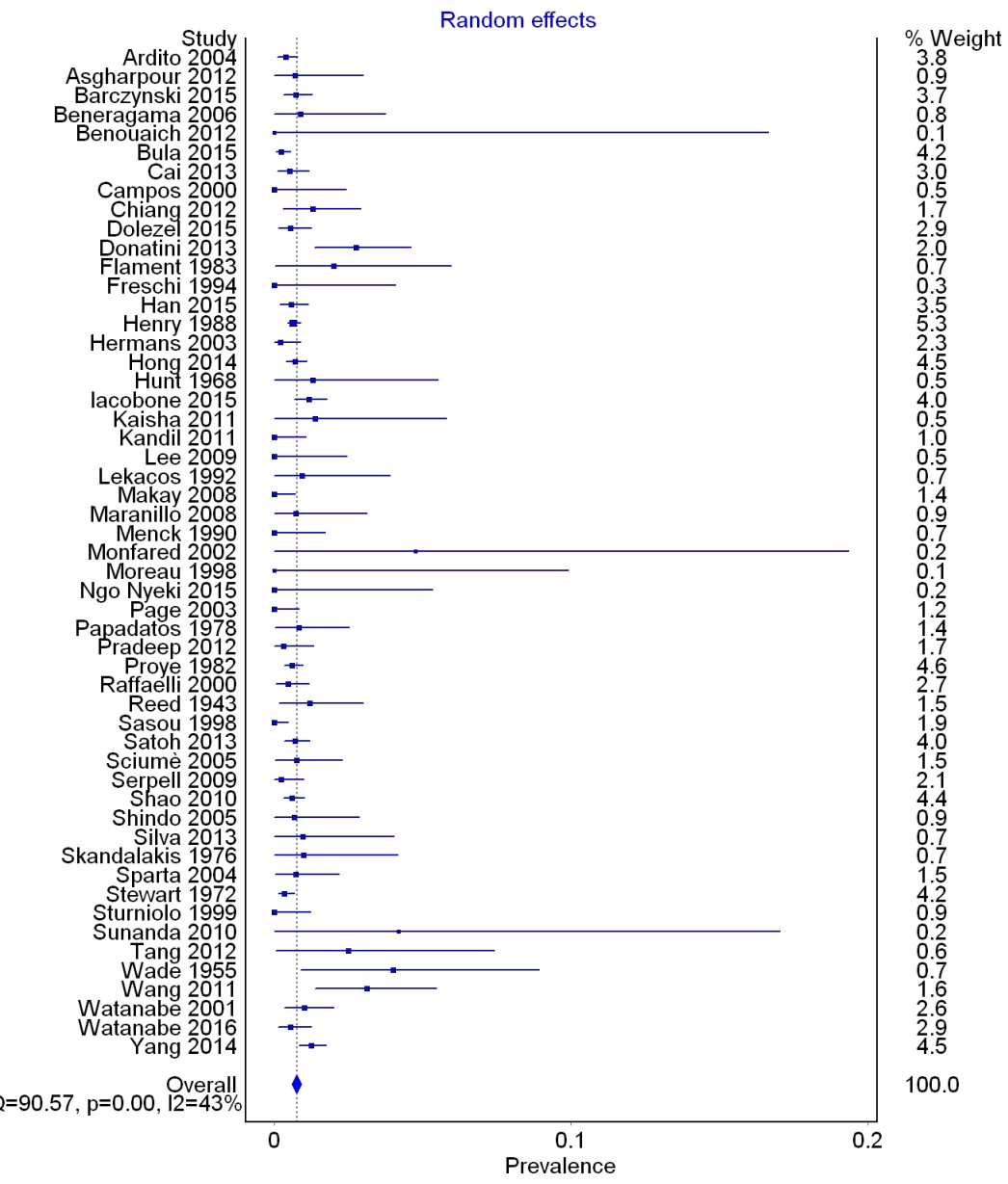

**Figure 4** Forest plot for pooled prevalence of a non-recurrent laryngeal nerve.

## Prevalence of an aberrant subclavian artery in cases of right NRLNs

In 21 studies with right NRLNs ($n = 136$ nerves), an aberrant subclavian artery was reported. An aberrant subclavian artery was present in 89.3% (95% CI [79.6–96.3]) of cases ($I^2 = 49.1\%$, 95% CI [15.7–69.3]; $p = 0.006$).

## Prevalence of a Left NRLN

A total of 41 studies with left RLNs ($n = 20,006$) reported data on the prevalence of a left NRLN. Only one of the included studies, reported the presence of left NRLNs (two cases,

**Table 1  Table of included studies.**

| Study | Country | Type | # of right RLNs | # of NRLN (Prevalence) |
|---|---|---|---|---|
| *Ardito et al. (2004)* | Italy | IP | 1342 | 5 (0.37%) |
| *Asgharpour et al. (2012)* | Spain | C | 143 | 1 (0.70%) |
| *Barczyński et al. (2015)* | Poland | IP | 1250 | 9 (0.72%) |
| *Beneragama & Serpell (2006)* | Australia | IP | 114 | 1 (0.88%) |
| *Benouaich et al. (2012)* | France | C | 10 | 0 (0%) |
| *Buła et al. (2015)* | Poland | IP | 1710 | 4 (0.23%) |
| *Cai et al. (2013)* | China | IP | 783 | 4 (0.51%) |
| *Campos & Henriques (2000)* | Brazil | C | 71 | 0 (0%) |
| *Chiang et al. (2012)* | Taiwan | IP | 310 | 4 (1.29%) |
| *Dolezel et al. (2015)* | Czech Republic | IR | 725 | 4 (0.55%) |
| *Donatini, Carnaille & Dionigi (2013)* | France | IR | 402 | 11 (2.74%) |
| *Flament, Delattre & Palot (1983)* | France | C | 100 | 2 (2%) |
| *Freschi et al. (1994)* | Italy | IP | 42 | 0 (0%) |
| *Han, Bai & Lu (2015)* | China | IR | 1056 | 6 (0.57%) |
| *Henry et al. (1988)* | France | IR | 4921 | 31 (0.63%) |
| *Hermans et al. (2003)* | Belgium | IP | 484 | 1 (0.21%) |
| *Hong, Park & Yang (2014)* | Korea | IR | 2187 | 15 (0.69%) |
| *Hunt, Poole & Reeve (1968)* | Australia | C | 77 | 1 (1.30%) |
| *Iacobone et al. (2015)* | Italy | IP | 1477 | 17 (1.15%) |
| *Kaisha, Wobenjo & Saidi (2011)* | Kenya | C | 73 | 1 (1.37%) |
| *Kandil et al. (2011)* | USA | IP | 162 | 0 (0%) |
| *Lee et al. (2009)* | Korea | C | 70 | 0 (0%) |
| *Lekacos et al. (1992)* | Greece | IR | 109 | 1 (0.92%) |
| *Makay et al. (2008)* | Turkey | IP | 250 | 0 (0%) |
| *Maranillo et al. (2008)* | Spain | C | 137 | 1 (0.73%) |
| *Menck, Grüber & Lierse (1990)* | Germany | C | 101 | 0 (0%) |
| *Monfared, Gorti & Kim (2002)* | USA | C | 21 | 1 (4.76%) |
| *Moreau et al. (1998)* | France | C | 17 | 0 (0%) |
| *Ngo Nyeki et al. (2015)* | Switzerland | IP | 32 | 0 (0%) |
| *Page, Foulon & Strunski (2003)* | France | IP | 205 | 0 (0%) |
| *Papadatos (1978)* | France | C | 239 | 2 (0.84%) |
| *Pradeep, Jayashree & Harshita (2012)* | India | IR | 324 | 1 (0.31%) |
| *Proye et al. (1982)* | France | IR | 2490 | 15 (0.60%) |
| *Raffaelli, Iacobone & Henry (2000)* | France | IP | 656 | 3 (0.46%) |
| *Reed (1943)* | USA | C | 253 | 3 (1.19%) |
| *Sasou, Nakamura & Kurihara (1998)* | Japan | IR | 367 | 0 (0%) |
| *Satoh et al. (2013)* | Japan | IR | 1561 | 11 (0.70%) |
| *Sciumè et al. (2005)* | Italy | IR | 263 | 2 (0.76%) |
| *Serpell, Yeung & Grodski (2009)* | Australia | IP | 432 | 1 (0.23%) |
| *Shao et al. (2010)* | China | IP | 1988 | 12 (0.60%) |
| *Shindo, Wu & Park (2005)* | USA | IP | 149 | 1 (0.67%) |

**Table 1** (*continued*)

| Study | Country | Type | # of right RLNs | # of NRLN (Prevalence) |
|---|---|---|---|---|
| *Silva, Siqueira & Arruda (2013)* | Brazil | C | 106 | 1 (0.94%) |
| *Skandalakis et al. (1976)* | USA | C | 102 | 1 (0.98%) |
| *Spartà et al. (2004)* | France | IP | 274 | 2 (0.73%) |
| *Stewart, Mountain & Colcock (1972)* | England | IP | 1776 | 6 (0.34%) |
| *Sturniolo et al. (1999)* | Italy | IR | 141 | 0 (0%) |
| *Sunanda, Tilakeratne & De Silva (2010)* | Sri Lanka | IP | 24 | 1 (4.17%) |
| *Tang et al. (2012)* | China | C | 80 | 2 (2.50%) |
| *Wade (1955)* | England | C | 100 | 4 (4%) |
| *Wang et al. (2011)* | China | IR | 290 | 9 (3.10%) |
| *Watanabe et al. (2001)* | Japan | Imaging (CT) | 594 | 6 (1.01%) |
| *Watanabe et al. (2016)* | Japan | IP | 730 | 4 (0.55%) |
| *Yang et al. (2014)* | China | IR | 2251 | 28 (1.24%) |

**Notes.**
RLN, Recurrent Laryngeal Nerve; NRLN, Non-Recurrent Laryngeal Nerve; C, Cadaveric; IP, Intraoperative Prospective; IR, Intraoperative Retrospective.

**Table 2** Type of study subgroup analysis for prevalence of an NRLN.

| | # of studies (# of nerves) | Prevalence of NRLN: % (95% CI) | $I^2$: % (95% CI) | Cochrane's Q, *p*-value |
|---|---|---|---|---|
| Overall | 53 (33571) | 0.7 (0.6–0.9) | 42.5 (20.2–58.6) | 0.001 |
| Cadaveric | 17 (1700) | 1.4 (0.9–2.0) | 0 (0–33.1) | 0.761 |
| Intraoperative | 35 (31277) | 0.7 (0.5–0.8) | 50.6 (27.1–66.5) | <0.001 |
| Intraoperative (Prospective) | 21 (14190) | 0.5 (0.4–0.7) | 22.7 (0-54.6) | 0.170 |
| Intraoperative (Retrospective) | 14 (17087) | 0.8 (0.6–1.1) | 64.9 (38.0–80.1) | <0.001 |

**Notes.**
NRLN, Non-Recurrent Laryngeal Nerve.

**Table 3** Geographical subgroup analysis for prevalence of an NRLN.

| | # of studies (# of nerves) | Prevalence of NRLN: % (95% CI) | $I^2$: % (95% CI) | Cochrane's Q, *p*-value |
|---|---|---|---|---|
| Overall | 53 (33571) | 0.7 (0.6–0.9) | 42.5 (20.2–58.6) | 0.001 |
| Africa | 2 (105) | 1.5 (0.0–4.3) | 0.0 (0.0–0.0) | 0.613 |
| Asia | 15 (12615) | 0.8 (0.6–1.1) | 54.2 (18.0–74.4) | 0.006 |
| Europe | 25 (17588) | 0.7 (0.5–0.9) | 45.0 (11.9–65.6) | 0.008 |
| North America | 6 (2463) | 0.7 (0.2–1.4) | 38.4 (0.0–75.5) | 0.149 |
| Oceania | 3 (623) | 0.6 (0.0–1.5) | 14.1 (0–91.1) | 0.312 |
| South America | 2 (177) | 0.9 (0.0–2.5) | 0.0 (0.0–0.0) | 0.449 |

**Notes.**
NRLN, Non-Recurrent Laryngeal Nerve.

both in patients with situs inversus), which equated to a pooled prevalence estimate of 0% (95% CI [0–0.1]; $I^2 = 0$%, $p = 1.0$) (*Henry et al., 1988*).

# DISCUSSION

The NRLN, a rare, often developmentally-derived variant of the RLN, most often results from partial failure of the pharyngeal apparatus during embryo development (*Watanabe et al., 2016*). An NRLN can very easily be injured surgically and this leads to long-term

postoperative complications such as vocal cord paralysis. Adequate identification and isolation is most important for preventing injury (*Toniato et al., 2004*). The frequency of NRLN injury remains poorly reported, vague, and believed to be continually underestimated (*Dolezel et al., 2015*).

The pooled prevalence rates of NRLN were calculated solely from studies that provided information about the rate per side in patients or cadavers. No NRLN has ever been noted on the left side in the absence of rare pathologies such as situs inversus with accompanying aortic arch abnormalities (*Toniato et al., 2004*; *Hong, Park & Yang, 2014*). To include studies that mixed right and left sides into one rate would dilute and thereby falsify the overall pooled prevalence rates. We therefore infer that the best representation of overall NRLN prevalence is its existence on the right side. Thus, the pooled prevalence estimate of a right-sided NRLN is a proxy of the pooled prevalence estimate of NRLN per person/cadaver. In support of our decision to include only studies that reported rates per side, we calculated the prevalence of left NRLNs on the basis of literature data. The prevalence was 0% in a sample of 20,006 left nerves examined, indicating that this anomaly occurs in <0.1% of the population.

We found an overall pooled prevalence of right NRLN of 0.7% in the general population. Subgroup analysis based on study modality revealed significant differences, NRLNs being found more than twice as often in cadavers as in operative subjects. We should note that because of such limitations in the intraoperative viewing of anatomical structures as equipment obstruction, edema, inflammation, and the small caliber of nerve branches, the cadaveric rate (1.4%) could reflect the NRLN's true prevalence better. However, the cadaveric group (1,700 nerves) was limited by its small sample size, dwarfed by the intraoperative group (31,277 nerves). Further subgroup analysis based on the geographical origin of the study revealed no notable differences among populations.

A subanalysis of variant nerves allowed the types of origins of the NRLN to be assessed on the basis of whether they lay above or below the laryngotracheal junction (LTJ); the prevalence values were 58.3% and 41.7%, respectively. Many previous articles have developed classification systems for NRLN origins, but very few have used the same system, the majority just describing the NRLNs identified (*Stewart, Mountain & Colcock, 1972*; *Henry et al., 1988*; *Toniato et al., 2004*; *Chiang et al., 2012*; *Hong, Park & Yang, 2014*; *Dolezel et al., 2015*). However, the nerves differed in their courses despite originating from similar levels on the vagus nerve. Some NRLNs exhibited a course in which the nerve immediately tracks medially and enters the larynx. As described in the study by *Toniato et al. (2004)*, most patients with NRLNs who experienced injuries in their series had nerves that originated above the LTJ and coursed with the superior thyroid artery. Another subset of patients had NRLNs originating above the LTJ but displaying a looping course, where after originating they descended inferiorly and then reascended superiorly before entering the larynx (Fig. 5) (*Toniato et al., 2004*; *Hong, Park & Yang, 2014*).

The prevalence of a right NRLN was strongly associated with the presence of aberrant subclavian artery, the causative anomaly of Dysphagia Lusoria (Bayford-Autenrieth Dysphagia) (*Watanabe et al., 2001*). The symptoms associated with an aberrant subclavian artery are very often silent, but if present can include dysphagia, chronic cough, and

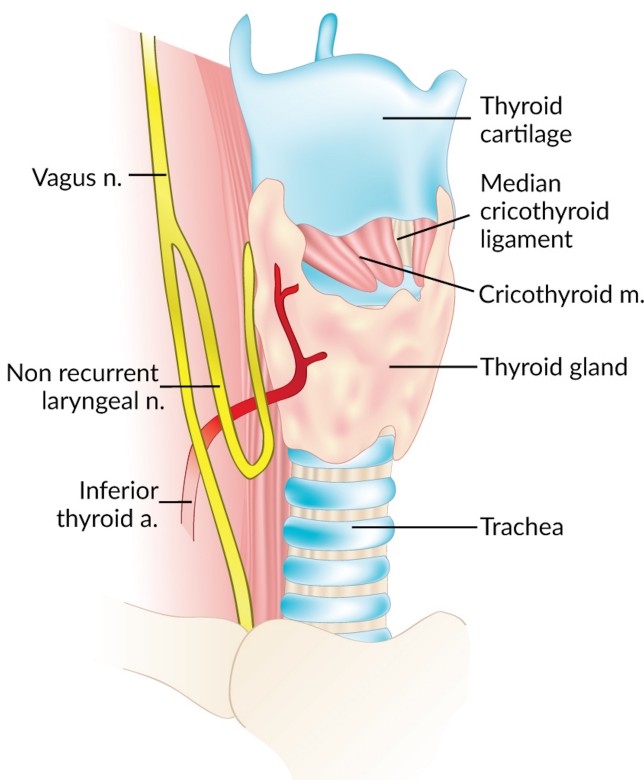

**Figure 5** Looping course of a right non-recurrent laryngeal nerve.

unexplained ischemia of the right upper limb (*Natsis et al., 2016*). We noted that 86.7% of right NRLN patients had an aberrant subclavian artery pattern. The embryological pathogenesis of an NRLN with an aberrant subclavian artery is clear and understood, but the presence of the variant nerve without the accompanying vascular anomaly remains a mystery (*Dolezel et al., 2015*). *Wang et al. (2011)* noted, along with the research by *Raffaelli, Iacobone & Henry (2000)*, that the origin of the NRLN was never confirmed to be the vagus in these non-aberrant subclavian artery cases. It is posited that the connecting branches between the sympathetic trunk and normal RLN could be mistaken for an NRLN (*Raffaelli, Iacobone & Henry, 2000*).

Preoperative identification of aberrant subclavian arteries and NRLNs is the best defense for a surgeon. As was noted by Iacobone for USG and Watanabe for Computed Tomography (CT), identification of these variant structures is potentially 100% of the time by using these imaging techniques (*Iacobone et al., 2015*; *Watanabe et al., 2016*). Another option for identification intraoperatively is the use of nerve monitoring (IONM) techniques. Dolezel reports that the use of IONM increased the prevalence of NRLNs yet decreased the incidence of postoperative nerve palsy (*Dolezel et al., 2015*). The use of IONM is particularly advantageous when patients have an underlying pathology which may restrict surgical dissection and viewability of the neural structures (*Barczyński et al., 2014*). The

IONM technology is continually becoming more advanced and provides a promising tool for use in future procedures.

Additional research on this topic is necessary to assess the possible etiologies of the NRLN when it occurs in the absence of an aberrant subclavian artery. Furthermore, morphometric analysis of the NRLN with regard to its origin and course would provide useful insight into its behavior and enable its location to predicted more readily for operative planning. Nonetheless, since the variant occurs in nearly 1% of the population and is associated with a high risk of iatrogenic injury, we recommend preoperative USG examination for all patients undergoing procedures in the anterior neck.

This study was limited by several facets, particularly, several studies were omitted from the meta-analysis due to the lack of reported data on the side of occurrence of the NRLN. Moreover, although we performed detailed subgroup investigations, high levels of heterogeneity lingered between the included studies. We suspect that this is because of inherent variability in the occurrence of the NRLN. Lastly, no quality and risk-of-bias assessments of included studies were performed due to a lack of an available tool for the field of anatomy.

## CONCLUSIONS

The NRLN is an asymptomatic and most often embryologically-derived variant of the RLN in which the nerve arises directly from the vagus at a cervical level. In healthy patients the anomaly is restricted to the right side and, if present, is a very clinically relevant structure, particularly for surgeons conducting procedures directly or requiring access to the anterior neck. Non-Recurrent Laryngeal Nerves are associated with increased risks for iatrogenic surgical injury, most often leading to either temporary or permanent vocal cord paralysis. While an NRLN occurs in only about 1% of the population, the high risk of iatrogenic injury indicates that its possible occurrence in a patient should be screened preoperatively using USG. A thorough and complete understanding of the prevalence, origin, and associated pathologies is vital for preventing injuries and for ensuring patient safety and operative success.

## ACKNOWLEDGEMENTS

Krzysztof A. Tomaszewski was supported by the Foundation for Polish Science (FNP) which recognizes the achievements of top scientists in the country. We wish to thank Karolina Saganiak for the anatomical drawings used in this manuscript.

### Funding

The publication of this manuscript was supported by the Faculty of Medicine, Jagiellonian University Medical College KNOW (Leading National Research Centre 2012–2017) funds. The funders had no role in study design, data collection and analysis, decision to publish, or preparation of the manuscript.

## Grant Disclosures

The following grant information was disclosed by the authors:

Faculty of Medicine, Jagiellonian University Medical College KNOW (Leading National Research Centre 2012–2017).

## Competing Interests

The authors declare there are no competing interests.

## Author Contributions

- Brandon Michael Henry conceived and designed the experiments, performed the experiments, analyzed the data, contributed reagents/materials/analysis tools, wrote the paper, prepared figures and/or tables, reviewed drafts of the paper.
- Silvia Sanna conceived and designed the experiments, performed the experiments, analyzed the data, prepared figures and/or tables, reviewed drafts of the paper.
- Matthew J. Graves and Beatrice Sanna performed the experiments, wrote the paper, prepared figures and/or tables, reviewed drafts of the paper.
- Jens Vikse performed the experiments, analyzed the data, prepared figures and/or tables, reviewed drafts of the paper.
- Iwona M. Tomaszewska analyzed the data, contributed reagents/materials/analysis tools, reviewed drafts of the paper.
- R. Shane Tubbs analyzed the data, reviewed drafts of the paper.
- Jerzy A. Walocha conceived and designed the experiments, contributed reagents/materials/analysis tools, reviewed drafts of the paper.
- Krzysztof A. Tomaszewski conceived and designed the experiments, analyzed the data, contributed reagents/materials/analysis tools, reviewed drafts of the paper.

## Data Availability

The raw data has been supplied as a Supplementary File.

## Supplemental Information

Supplemental information for this article can be found online at http://dx.doi.org/10.7717/peerj.3012#supplemental-information.

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
