# Peer review of "The Non-Recurrent Laryngeal Nerve: a meta-analysis and clinical considerations"

_PeerJ, doi:10.7717/peerj.3012_

## Round 0.1 · original submission · Major Revisions

· Academic Editor

Major Revisions

As you will see, two of the reviewers offered no context for their recommendation. Reviewers #3 and #4 have raised some methodological issues that should be addressed to their satisfaction. Reviewer #4 also offers a number of studies to be considered, and raises the need to explore the literature more broadly, which I hope you will address to the reviewer's satisfaction.

·

Basic reporting

No comments

Experimental design

No comments

Validity of the findings

No comments

Additional comments

I wish to congratulate the authors for their valuable work. Accept without any comment.

·

Basic reporting

Interesting and clinically relevant topic never before studied on this scale.

Experimental design

PRISMA methodology.

Validity of the findings

Valid based on number of articles and respective patients reported.

Additional comments

A nice review.

Reviewer 3 ·

Basic reporting

No comments

Experimental design

No comments

Validity of the findings

In Supplemental Table 3 some other information on studies should be reported for example mean age of patients.

Statistical methods are clear. Just one integration should be made: in the subgroup analysis to statistically test differences between subgroups (i.e.: cadaveric vs intraoperative) a Cochrane' s Q test for interaction should be used and p-value reported when comparing the prevalence rates (i.e: 1.4% vs 0.7%). The single subgroup p-values as reported in table 2 or table 3 have no relevance without an interaction test.

Drop all results from discussion (especially confidence intervals that should be reported only in results section).

Reviewer 4 ·

Basic reporting

No Comments

Experimental design

No Comments

Validity of the findings

No Comments

Additional comments

General overview:
Interesting paper on the frequency of the Non-Recurrent Laryngeal Nerve as a rare anatomical variant.
However, I have some comments for the authors; mainly the need to check old references and non-English databases.
1. In the paragraph ‘’Data extraction’’, the authors should indicate and define separately the primary outcome (prevalence of NRLN) and the secondary outcomes (level of origin of the NRLN, prevalence of an aberrant subclavian artery when a NRLN was present).
2. In the paragraph ‘’statistical analysis’’ the authors mentioned: For the Chi2 test, a Cochran’s Q p-value of <0.10 indicated significant heterogeneity. In fact, there is no need to report the Q-Cochrane test neither its P value.
3. In same paragraph, the authors stated ‘’ All pooled prevalence rates were calculated using a random effects model’’. Why? If I2 value is less than 50%, it is usually admitted to report the fixed-effect model estimate. The new classification offered by Cochrane Handbook for Systematic Reviews of Interventions, meant for comparative interventions, is meant to be just a rough guide to I2 interpretation.
4. In the paragraph ‘’ Study Identification’’, the authors should remove duplicates of the database search results before reference checking.
5. There is no need to report the CI of the I2 statistic value.
6. In the discussion, first sentence ‘’ The NRLN, a rare, often developmentally-derived variant of the RLN, most often results from partial failure of the pharyngeal apparatus during embryo development’’, needs referencing.
7. It is said ‘’ the prevalences were 58.3% and 41.7%, respectively.’’ It should be written the prevalence values…
8. I believe that the study of watanabe et al. should be removed: its primary outcome was to use CT to locate a type 1 ‘’aberrant’’ subclavian artery and not a NRLN; and they found 4 with such type out of 730 patients, and only these 4 patients had a surgical exploration for a NRLN. These authors didn’t search for a NRLN in the remaining 726 patients.
9. The study of Reeves et al. could be included (Reeve TS, Coupland GAE, Johnson DC, et al: The recurrent and external laryngeal nerves in thyroidectomy. Med J Aust 1:380-382, 1969). These authors, after a recent review of their personal thyroidectomies, found seven such nerves in 1,200 operations. Interestingly, a nonrecurrent laryngeal nerve has never been found on the left side.
10. Koumare et al. Nerf Laryngé Inférieur : anatomie et lésions opératoires. e-mémoires de l'Académie Nationale de Chirurgie, 2002, 1 (2) : 8-11 ; could be included (they found the NRLN in 1.4%; 16/1133 operative cases).
11. Campus and Henriques. RELATIONSHIP BETWEEN THE RECURRENT LARYNGEAL NERVE AND THE INFERIOR THYROID ARTERY: A STUDY IN CORPSES. REV. HOSP. CLÍN. FAC. MED. S. PAULO 55(6):195-200, 2000. Might be included as well.
12. I beleive you should check the following papers (BLONDEAU P. Rapports chirurgicaux du nerf récurrent et de l’artère thyroïdienne inférieure. Journal de Chirurgie. 1971 ; 102 : 397-414. BLONDEAU P, LEDUCQ J, RENE L. Plaidoyer pour la dissection complète du nerf récurrent dans la lobectomie thyroïdienne totale. Mém. Acad. Chir. 1971 ; 97 : 446-58. BLONDEAU P, LEDUCQ J, ROULLEAU P, RENE L. Les risques fonctionnels de la chirurgie thyroïdienne. Etude d’une série de 1000 interventions. Le risque récurrentiel. Ann. Chir.1973 ; 27 :771-80. BLONDEAU P, NEOUZE L, RENE L. Le nerf laryngé inférieur non récurrent, danger de la chirurgie thyroïdienne. (7 observations). Ann. Chir. 1977 ; 31 : 917-23.
It seems to me that more references still could be found. Checking the French and the Latino American literature could be beneficial in locating more relevant papers.

13. Holzapfel Gotthold. Ungewohnlicher Ursprung und Verlauf der Arteria Subclavia Dextra. Anatomische Hefte, I Abt, bd. Xii, p 369, 1899. His paper is interesting in showing the relationship of NRLN with an anomalous subclavian artery: 3/35 recurrence of the artery, 4/35 recurrence around the right vertebral artery (which in this case was a branch of the right carotid artery), and the remaining 28/35 were non-recurrent.
14. Williams (1933) has examined 159 aortic arches and four showed anomalous subclavian artery variations with a direct inferior laryngeal nerve. (Williams GD. ANOMALY OF THE INFERIOR LARYNGEAL NERVE. Annals of Surgery 97(6) June 1933.
15. PRISMA checklist is meant for interventional comparative studies. I would suggest the authors to use the CARMA checklist meant specifically for anatomical meta-analysis. K. Yammine. Evidence-based Anatomy in Clinical Anatomy. 2014

---

## Round 0.2 · Major Revisions

· Academic Editor

Major Revisions

Thank you for addressing the changes suggested after your original submission. As you will see, while one one of the reviewers is satisfied with the changes you have made, Reviewer #4 has identified some remaining issues. I will ask you to go through these, and, if I may, suggest that those are addressed in the body of the manuscript. In other words, making those changes which you feel are justified, and addressing, explicitly those where differences in opinion between the reviewer and your group's remain. For example, I see there is continued disagreement as to whether I2 and CI should all be reported, or whether CARMA or PRISMA is the best approach, etc. I am sure readers may have those thoughts as well, and it would be beneficial to have a justification of your choices explicitly addressed. These points of view could be easily incorporated either as part of your methods section or in the discussion, with some context regarding the pros/cons associated with your and the reviewer's approach, and this should expedite the processing of the article.

Reviewer 3 ·

Basic reporting

No comments

Experimental design

No comments

Validity of the findings

No comments

Additional comments

Authors answered sufficiently to the criticisms observed in the original version.

Reviewer 4 ·

Basic reporting

Please refer to the general comments

Experimental design

Please refer to the general comments

Validity of the findings

Please refer to the general comments

Additional comments

Second revision comments
The replies the authors proposed over Cochrane Q and I2 statistic are too technical for an audience of ‘’anatomists’’’ and not considered as straightforward as the authors are implying. On the contrary:
- Heterogeneity is known to be high in prevalence anatomical studies, we look for a simple measure to test it; I2 value could give an appropriate and simple estimate for interpretation. Besides, random effect model estimate is not the ideal one to be used when the number of studies is low.
- Per Cochrane group, ‘’The rough guide to interpretation is as follows:
0% to 40%: might not be important;
30% to 60%: may represent moderate heterogeneity*;
50% to 90%: may represent substantial heterogeneity*;
75% to 100%: considerable heterogeneity*.’’

As stated by Cochrane it’s a rough guide (I would say rough and vague); what is your interpretation of a I2 of 75%, it indicates a ‘’substantial’’ or ‘’considerable’’ heterogeneity? In its simplest and practical form: when I2 is above 50%, heterogeneity is to be considered high and the opposite is true.

- Reporting the CI of I2 along with the p-value of Q in MA of observational studies is very confusing: the CI of I2 value is always positive as you can see! What does it mean the following results in terms of heterogeneity?
‘’Prevalence of a Right NRLN: 0.7% [95%CI: 0.6-0.9; I2 =42.5 (95%CI: 20.2-58.6); p=0.001]
Types of Right NRLN: (I2=67.6%, 95%CI: 43.4-81.4; p< 0.001).
Prevalence of an aberrant subclavian artery: (I2=49.1%, 95%CI: 15.7-69.3; p=0.006).
I2 values along with non-zero-included CI ranges were associated to p-value of Q which was largely less than 0.05 of significance; does it mean that heterogeneity is considerable for all 3 outcomes?

I believe that is very confusing and would again ask the authors to only use the I2 (and P-values if they wish) values but without the CI of I2.

- Since the Prevalence of a Left NRLN is quite nil (meta-analytical prevalence = 0%), the exclusion criteria of no reporting side prevalence values is no more valid. All suggested studies reporting overall NLRN prevalence could be included; getting a bigger pooled sample would yield a more accurate estimate of this variant since it is a very rare one.

- Additionally, I don’t believe that a study should be automatically excluded if it did not report NRLN as a study endpoint; a study may have multiple endpoints. perhaps the authors meant a primary endpoint; however, if a study reported a prevalence value as a secondary outcome, it could be included even if its not a primary endpoint. Such is the case of many retrospective surgical studies.

- Geographical variable is different from ancestry variable; the ancestry should be reported not the geographical area, when available or highly suspected. Only otherwise, geography is reported.

- Lastly, CARMA checklist is the result of combining 3 checklists including PRISMA; what you may find in PRISMA is already included in CARMA. More, it includes the ‘’observational’’ items of two other valid checklists; MOOSE and STROBE, and where such items were specifically adapted to the anatomy field. Using a checklist meant for interventional studies while 3 suitable checklists are already published and tested is difficult to grasp. However, the decision is left to the authors; i would recommend a brief explanation/rationale of why using the PRISMA for a MA of observational studies.

---

## Round 0.3 · accepted · Accept

· Academic Editor

Accept

Thank you for your patience as I consulted on this decision. I am happy that I am able to accept this manuscript on behalf of PeerJ.